# Association of Hematuria with Renal Progression and Survival in Patients Who Underwent Living Donor Liver Transplant

**DOI:** 10.3390/jcm10194345

**Published:** 2021-09-24

**Authors:** Kai-Chieh Chang, Yao-Peng Hsieh, Huan-Nung Chao, Chien-Ming Lin, Kuo-Hua Lin, Chun-Chieh Tsai, Chia-En Heish, Pei-Ru Lin, Chew-Teng Kor, Yao-Li Chen, Ping-Fang Chiu

**Affiliations:** 1Division of Nephrology, Department of Internal Medicine, Changhua Christian Hospital, Changhua 50006, Taiwan; 200648@cch.org.tw (K.-C.C.); 102407@CCH.ORG.TW (Y.-P.H.); 1310008@cch.org.tw (H.-N.C.); 719696@cch.org.tw (C.-M.L.); 144942@cch.org.tw (C.-C.T.); 2School of Medicine, Kaohsiung Medical University, Kaohsiung 807378, Taiwan; 3Division of Nephrology, Hanming Christian Hospital, Changhua 50058, Taiwan; 4Department of General Surgery, Changhua Christian Hospital, Changhua 50006, Taiwan; 120380@cch.org.tw (K.-H.L.); 69918@cch.org.tw (C.-E.H.); 5Big Data Center, Changhua Christian Hospital, Changhua 50006, Taiwan; 183778@cch.org.tw (P.-R.L.); 179297@cch.org.tw (C.-T.K.); 6Transplant Medicine & Surgery Research Centre, Changhua Christian Hospital, Changhua 50006, Taiwan; 7School of Medicine, Chung Shan Medical University, Taichung 40201, Taiwan; 8Department of Hospitality Management, Ming Dao University, Changhua 52345, Taiwan

**Keywords:** liver transplantation, hematuria, chronic kidney disease, survival

## Abstract

Background: This study aimed to determine the association between episodic or persistent hematuria after liver transplantation and long-term renal outcomes. Methods: Patients who underwent living donor liver transplantation between July 2005 and June 2019 were recruited and divided into two groups based on the finding of microscopic or gross hematuria after transplantation. All patients were followed up from the index date until the end date in May 2020. The risks of chronic kidney disease, death, and 30% and 50% declines in estimated glomerular filtration rate (eGFR) were compared between groups. Results: A total of 295 patients underwent urinalysis for various reasons after undergoing transplantation. Hematuria was detected in 100 patients (group A) but was not present in 195 patients (group B). Compared with group B, group A had a higher risk of renal progression, including eGFR decline >50% [aHR = 3.447 (95%CI: 2.24~5.30), *p* < 0.001] and worse survival. In addition, patients who took non-steroidal anti-inflammatory drugs (NSAIDs) continuously for over seven days within six months before transplant surgery had high risks of rapid renal progression, including a >30% decline in eGFR [aHR = 1.572 (95%CI: 1.12~2.21), *p* = 0.009)]. Conclusion: Development of hematuria after surgery in patients who underwent living donor liver transplant and were exposed to NSAIDs before surgery were associated with worse long-term renal dysfunction and survival.

## 1. Introduction

Hematuria is a sign of various conditions of kidney disease and can be of glomerular or non-glomerular origin. The common causes of glomerular hematuria include systemic lupus erythematosus, vasculitis, IgA nephropathy, and thin basement membrane nephropathy [1]. Non-glomerular hematuria can originate from the urinary system, which comprises the renal pelvis, ureter, and bladder. In men, the prostate gland communicates with the urinary system. Cancer, infection, and urolithiasis are common causes of non-glomerular hematuria. Several methods help distinguish between glomerular and non-glomerular hematuria; these include the presence of dysmorphic red blood cells or nephrotic proteinuria. However, regardless of its origin, hematuria has been reported to influence patient outcomes and renal function in the general population. Hematuria is associated with the incidence of acute kidney injury (AKI) [2] and chronic kidney disease (CKD) [3]. Moreover, a long-term follow-up study revealed that persistent asymptomatic hematuria correlated with increased risk of renal progression [4]. In patients who underwent kidney transplantation, the frequent incidence of urothelial malignancy may be attributed to an immunocompromised status and the underlying etiology of end-stage renal disease (ESRD). On the other hand, in adult patients who have undergone liver transplantation, no study has focused on the effects of hematuria on renal outcomes. Accordingly, we conducted the present study to investigate the effects of hematuria on urinalysis and renal outcomes.

## 2. Materials and Methods

Patients who underwent living donor liver transplant at Changhua Christian Hospital between July 2005 and June 2019 were enrolled. These patients had either liver failure or hepatocellular carcinoma that fulfilled the Milan criteria for liver transplantation. The index date was defined as the date of achieving stable renal function after surgery, around three months after transplantation. Patients who died during the hospitalization for transplantation and those who developed AKI immediately or within a short period after surgery and recovered with an estimated glomerular filtration rate (eGFR) of <90% were also excluded. This study was approved by the institutional review board of Changhua Christian Hospital, Taiwan (IRB 210131).

The enrolled patients were divided into two groups: those who had persistent or episodic hematuria on their urinalyses after the index date (group A) and those who did not have hematuria (group B). The diagnostic criteria for hematuria were defined as five or more red blood cells/HPF (high-power field, X 400) in the urinary sediment sample. The following variables were compared between the groups: age; comorbidity; previous exposure to medications; and laboratory data, including complete blood count, albumin level, baseline liver and renal function test results, and electrolytes. All patients were followed up from the index date until the end of May 2020. In addition, the risks of CKD (eGFR < 60 mL/min/1.73 m^2^) occurrence, death, and eGFR decline by 30% or 50% were compared between groups. The endpoints, including 30% and 50% decline of eGFR, were defined as surrogate outcomes, because few patients progressed to ESRD. Both surrogate endpoints that were applied simultaneously were practical for the enrolled patients who presented with varying renal function trajectories, especially the acute on CKD pattern [5]. In addition, the risk factors for the renal outcomes were evaluated. Sensitivity analysis was conducted for subgroups of patients who had no CKD (eGFR ≥ 60 mL/min/1.73 m^2^) and those in whom the identified causes of hematuria were urinary tract infection (UTI) and urolithiasis.

Data are presented as mean ± standard deviation for continuous variables and as numbers (percentages) for categorical variables. Student’s *t*-test and χ^2^ test were used to compare the continuous and categorical variables, respectively, between the two groups. The survival curves of one of the outcomes (i.e., decline in eGFR over time) in groups A and B were estimated using the Simon and Makuch method, which is an alternative to the Kaplan–Meier estimation, because the cases in group A included a time-varying exposure. Mantel and Bayer’s tests was used to compare the survival curves between the groups. Considering the immortal time bias for group A and the competing risk of death, we used time-dependent Cox models to determine the association between infections and renal function decline. The results were reported as hazard ratios (HRs) with 95% confidence intervals (CIs). Confounders, which were the variables with *p* < 0.05, were selected on the basis of crude HRs that were adjusted for in the multivariate Cox analysis to estimate the adjusted HRs (aHRs). All statistical analyses were conducted using the statistical package SPSS (v20; IBM Corporation, Chicago, IL, USA). Values with two-sided *p* < 0.05 were considered statistically significant.

## 3. Results

In total, 295 patients met the eligibility criteria and were categorized into two groups according to the development of urinalysis-confirmed hematuria secondary to various causes after the index date: group A (*n* = 100) and group B (*n* = 195). Table 1 presents the following demographic characteristics of the patients in both groups: sex; age; comorbidities; baseline liver and renal function test results; and prescribed medications, including nonsteroidal anti-inflammatory drugs (NSAIDs) and angiotensin-converting enzyme inhibitors/angiotensin II receptor blockers (ACE-I/ARBs). The mean eGFR on the index date was 75.5 ± 32.8 mL/min/1.73 m^2^ in group A and 75.2 ± 25.8 mL/min/1.73 m^2^ in group B.

No significant differences between the groups were observed in terms of sex, Charlson comorbidity index score, diabetes, cirrhosis, coronary artery disease, congestive heart failure, and hepatitis B (Table 1). In addition, the number of patients who had continuous exposure to NSAIDs for over seven days within six months before surgery was comparable between the groups. The two groups exhibited similar eGFR, hemoglobin, platelet, albumin, electrolyte levels, and coagulation test results.

On five-year follow-up, the risk of a 50% decline in eGFR was 50 (50%) in group A and 49 (25.1%) in group B (*p* < 0.001) and that of a 30% decline in eGFR was 74 (74%) in group A and 108 (55.4%) in group B (*p* = 0.002). In addition, the interval of occurrence of renal outcomes was shorter in group A than in group B (0.7 ± 1.2 years vs. 1.3 ± 1.7 years). The development of hematuria was not associated with the incidence of liver allograft rejection during follow-up (9 (9%) vs. 11 (5.6%), *p* = 0.277; Table 1). At the end of observation, mortality was higher in group A than in group B (27 (27%) vs. 28 (14.4%), *p* = 0.008). (Table 1, Figure 1).

When using a Cox proportional hazard regression to determine the association between transplant procedures with or without infections and renal outcomes, after adjustment for confounders (Table 2, Table 3 and Table 4, and Figure 1), group A, compared with group B, had higher a risk of renal dysfunction (aHR: 1.939, 95% CI: 1.23–3.06, *p* = 0.005), >30% or >50% decline in eGFR (aHR: 3.447, 95% CI: 2.24–5.30, *p* < 0.001), CKD, and composite outcomes (aHR: 1.926; 95% CI: 1.11–3.51, *p* = 0.027). Moreover, women and patients with relatively low albumin levels at baseline had a relatively high risk of renal composite outcomes (Table 2). Patients who took NSAIDs continuously for over seven days within six months before surgery had a relatively high risk of rapid renal dysfunction progression, including a >30% decline in eGFR and renal composite outcomes. Similarly, patients who took ACE-I/ARBs continuously for over 28 days within six months before surgery had a relatively high risk of composite outcomes. However, the number of patients with NSAID exposure after liver transplant was comparable between both groups (27 (27%) vs. 48 (24.6%)). NSAID intake after surgery had less influence on the final renal function at the end of the observation period than NSAID intake before surgery (>30% or >50% decline in eGFR, *p* = 0.33 and 0.76, respectively, not shown).

Further analysis of the causes of hematuria revealed that 26 patients had renal stones and that 49 patients had episodes of UTI. Moreover, 17 patients developed both UTI and urolithiasis. Two patients had prostate hypertrophy. Long-term antiplatelet medications were prescribed for 17 patients with coronary artery or cerebrovascular disease, and anticoagulants were used to treat atrial fibrillation in three patients. A total of 19 patients with UTI or urolithiasis received antiplatelet medication; however, no patient died from proven urinary tract malignancy. After completion of the study, we could not determine the etiology of hematuria in 31 patients (31%).

Table 5 shows the results of the sensitivity analysis of different subgroups according to infection, after excluding patients with eGFR of <60 mL/min/1.73 m^2^ (aHR: 2.913, 95% CI: 1.38–6.13; *p* = 0.005). In addition, we observed significant results after controlling for UTI or urolithiasis. Hematuria was independently associated with negative outcomes (aHR: 2.596, 95% CI: 1.04–6.49, *p* = 0.041; aHR: 2.86, 95% CI: 1.50–5.45, *p* < 0.001).

Further investigation focused on the influence of proteinuria in urinalysis on renal outcomes. Of 295 patients, 164 had proteinuria on urinalysis. Patients with incidental proteinuria with or without hematuria had more rapid progression of renal composite outcomes compared with that in patients who did not have proteinuria (aHR = 1.688, 95% CI: 1.04–2.75; *p* = 0.036).

## 4. Discussion

In addition to long-term administration of calcineurin inhibitors, diabetes mellitus, hypertension, and kidney dysfunction before liver transplant were the reported factors associated with CKD incidence following liver transplant [6]. In addition, the risk of CKD was reported to be significantly associated with advanced age, female sex, and hepatitis C carrier status before transplantation [7]. Lee et al. found that the overall risk of CKD (eGFR < 60 mL/min per 1.73 m^2^) correlated with low pretransplant eGFR values, pretransplant hepatorenal syndrome, pretransplant proteinuria levels, and higher Child–Pugh and MELD scores [8].

In a 22-year follow-up study, the presence of isolated microscopic hematuria was found to be associated with a significantly increased risk of ESRD4 in the general population. In addition, hematuria was associated with a significantly high risk of death in the first two years of follow-up [9]. Hematuria had been considered an indicator of activity of glomerular nephropathy in vasculitis, lupus nephritis, or IgA nephropathy [10,11,12]. In addition, persistent hematuria was reported to be associated with an increased risk of ESRD in IgA nephropathy [13]. The degree of proteinuria definitely correlates with the risk of diabetic kidney; however, microscopic hematuria is also a dependent risk factor for ESRD in diabetic nephropathy [14]. These suggest that the presence of nondiabetic renal disease in patients with diabetic CKD carries a relatively high risk of renal progression [15]. Furthermore, glomerular hematuria can lead to AKI, with the pathogenesis primarily involving bouts of gross hematuria, including intratubular obstruction of blood casts with consequent acute tubular necrosis, and direct toxic tubular effects of hemoglobin. In addition, heme may trigger oxidative stress and erythrophagocytosis through renal tubular cells1.

Non-glomerular hematuria includes UTI, urolithiasis, and prostatic disorders. Among liver transplant recipients, UTI in the first year was identified as an independent risk factor for CKD stage progression [16]. Several studies have consistently observed a relationship between a history of nephrolithiasis and a two-fold increase in the risk of both CKD and ESRD [17]. After stone formation in patients with CKD, the specific complications include obstructive uropathy, recurrent UTI, and struvite stones. Urolithiasis is managed through modalities, such as extracorporeal shock wave lithotripsy and ureterolithotripsy, which can further cause kidney injury. In part, this finding indicates that kidney stone formation is common in patients with hypertension, gout, and diabetes mellitus. In renal transplant patients, the incidence of renal allograft stones has been associated with reduced graft survival [18]. Frequent urinalysis in such patients enables early and easy recognition of obstructive uropathy. On the other hand, in patients who have undergone liver transplant, the diagnosis of obstructive uropathy is probably delayed.

Patients on antithrombotic agents may have increased incidence of gross hematuria [19]. Anticoagulants and their corresponding medical conditions are commonly referred to as anticoagulant-related nephropathy, which was found to be correlated with the progression of kidney dysfunction [20]. Moreover, the incidence of microscopic hematuria was found to be elevated in elderly patients who received regular doses of aspirin [21], a low dose aspirin may play a role in renal progression in patients with CKD [22]. However, contrasting results were observed in patients who were older [23] or had diabetes [24]. A further cohort study is ongoing to clarify this issue [25].

Rare etiologies of hematuria have been reported in patients who have received organ transplants. In a liver transplant recipient, hemorrhagic cystitis was reported to have developed secondary to BK virus infection [26]. In our study, although hematuria has several causes, its appearance was consistently shown to predict renal outcomes in patients who received living donor liver transplants. Routine urinalysis and early surveillance of the underlying disease in such patients were the essential tasks. In this study, the long-term renal outcomes were affected by hypoalbuminemia and continuous exposure to NSAIDs within seven days before surgery or to ACE-I/ARBs within 28 days before surgery, in addition to the hematuria-related disorders. Development of adverse renal outcomes with NSAID use after liver transplant is well known. However, exposure to NSAIDs or ACE-I/ARBs before the associated poor renal outcomes was an interesting finding. Perhaps patients with liver failure are always in a state of low effective intravascular volume. ACE-I/ARBs and NSAIDs may cause further deterioration of the renal injury. Male sex appeared to be a protective factor in patients who received a living donor liver transplant; this finding may be attributed to the higher mean eGFR on the index date in men than in women (77.3 ± 26.6 vs. 68.9 ± 32.5; *p* = 0.029). One review showed that asymptomatic hematuria in adults had unknown etiology in 43%–68% [27]; for these 31 patients (31%) without determinate etiology, regular and close follow-up every three to six months may be considered.

The current study had several limitations. First, this was a single-center study that enrolled a limited number of patients. Second, we were unable to further distinguish between gross and microscopic hematuria because of patient subjectivity. Quantitative analysis of proteinuria (proteinuria-creatinine ratio or 24 h urine sample) was not monitored routinely. Third, we surveyed surrogate outcomes instead of definite outcomes (i.e., death, ESRD, and renal transplant), because only few patients achieved these solid endpoints. Fourth, the data of deceased patients who had received liver transplant were not considered in this study because of the various conditions that may have been acquired from the donors.

## 5. Conclusions

The appearance of hematuria consistently predicted the renal outcomes and survival of patients who underwent living donor liver transplant. Routine urinalysis and early surveillance of the underlying disease is essential for such patients.

## Figures and Tables

**Figure 1 jcm-10-04345-f001:**
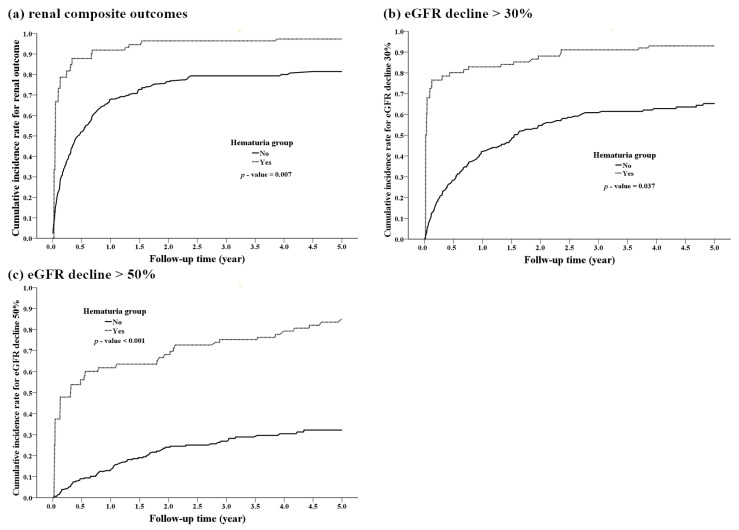
Patients with hematuria had rapid renal disease progression. (**a**) renal composite outcomes, (**b**) eGFR decline >30%, (**c**) eGFR decline >50%.

**Table 1 jcm-10-04345-t001:** Demographic data.

	Group A (Hematuria+)	Group B (Hematuria-)	*p*-Value
Patients (*n*)	100	195	
Age	55.1 ± 7.6	55.6 ± 7.9	0.555
Gender, Male	70 (70%)	154 (79%)	0.088
Comorbidity			
CCI	4.1 ± 4.2	4 ± 3.6	0.774
Diabetes mellitus	17 (17%)	31 (15.9%)	0.808
Hypertension	8 (8%)	24 (12.3%)	0.260
Hyperlipidemia	3 (3%)	10 (5.1%)	0.399
Hepatitis B	37 (37%)	71 (36.4%)	0.921
Hepatitis C	30 (30%)	44 (22.6%)	0.163
Cirrhosis	51 (51%)	98 (50.3%)	0.904
CHF	9 (9%)	14 (7.2%)	0.581
CAD	1 (1%)	3 (1.5%)	0.705
Medication before surgery			
NSAID	23 (23%)	39 (20%)	0.549
ACE-I/ARB	7 (7%)	16 (8.2%)	0.715
Laboratory data at surgery			
Hemoglobin (g/dL)	9.4 ± 1.6	9.8 ± 1.7	0.053
Albumin (g/dL)	2.86 ± 0.72	2.94 ± 0.69	0.326
AST (U/L)	179.0 ± 99.5	173.3 ± 106.1	0.653
ALT (U/L)	112.3 ± 59	115.3 ± 73.1	0.723
PT (second)	18.8 ± 4.9	17.9 ± 4.1	0.135
INR	1.7 ± 0.4	1.6 ± 0.4	0.231
APTT (second)	39.9 ± 11.2	37.5 ± 8.5	0.063
Platelet (10^3^)	84.6 ± 34.3	85.8 ± 42.7	0.798
Creatinine (mg/dL)	1.24 ± 0.79	1.22 ± 0.82	0.898
Laboratory data at index date			
BUN (mg/dL)	21.0 ± 16.9	17.1 ± 12	0.045
Creatinine (mg/dL)	1.10 ± 0.44	1.07 ± 0.36	0.569
eGFR (mL/min/1.73^2^)	75.5 ± 32.8	75.2 ± 25.8	0.930
Events after index date during follow-up			
Liver rejection	9(9%)	11(5.6%)	0.277
Average tacrolimus level (ng/mL)	5.3 ± 2.2	5.2 ± 1.5	0.785
^1^ NSAID exposure	27 (27%)	48 (24.6%)	0.656
Outcome after index date			
Mortality	27 (27%)	28 (14.4%)	0.008
eGFR decline > 30%	74 (74%)	108 (55.4%)	0.002
eGFR decline > 50%	50 (50%)	49 (25.1%)	<0.001
CKD (eGFR < 60)	65 (65%)	110 (56.4%)	0.155
Renal composite outcomes	88 (88%)	148 (75.9%)	0.014
Follow-up time			
Time to CKD (years)	1.4 ± 1.7	1.8 ± 1.8	0.046
Time to renal composite outcomes (years)	0.7 ± 1.2	1.3 ± 1.7	<0.001

^1^ NSAIDs were prescribed continuously for over seven days before surgery. SI: bloodstream infection; CCI: Charlson comorbidity index; CKD: chronic kidney disease; CHF: congestive heart failure; CAD: coronary artery disease. NSAIDs: nonsteroidal anti-inflammatory drugs; ACE-I/ARBs: angiotensin-converting enzyme inhibitor/angiotensin receptor blockers; BUN: blood urea nitrogen; AST: aspartate aminotransferase; ALT: alanine aminotransferase; PT: prothrombin time; aPTT: activated partial thromboplastin time; CRP: C-reactive protein

**Table 2 jcm-10-04345-t002:** Effects of hematuria on renal composite outcomes during a five-year follow-up.

	cHR (95% CI)	*p*-Value	aHR (95% CI)	*p*-Value
Hematuria	2.239 (1.30, 3.87)	0.004	1.926 (1.08, 3.44)	0.027
Age	1.021 (1.00, 1.04)	0.013	1.017 (1.00, 1.04)	0.062
Sex, Male	0.612 (0.46, 0.82)	0.001	0.681 (0.5, 0.93)	0.014
Diabetes mellitus	1.440 (1.03, 2.02)	0.035	1.193 (0.84, 1.69)	0.321
BUN	1.012 (1.00, 1.02)	0.002	1.01 (1.00, 1.02)	0.017
Albumin	0.783 (0.65, 0.95)	0.011	0.768 (0.62, 0.94)	0.012
NSAID (7 days)	1.505 (1.12, 2.03)	0.007	1.556 (1.15, 2.12)	0.005
ACE-I/ARB	1.77 (1.14, 2.75)	0.011	1.751 (1.11, 2.76)	0.016

Renal composite outcomes (30% or 50% decline in eGFR, CKD, or death). NSAIDs were prescribed continuously for over seven days before surgery; ACE-I/ARBs were prescribed continuously for over 28 days before surgery. CKD: chronic kidney disease; NSAIDs: nonsteroidal anti-inflammatory drugs; ACE-I/ARBs: angiotensin-converting enzyme inhibitor/angiotensin receptor blockers; BUN: blood urea nitrogen.

**Table 3 jcm-10-04345-t003:** Effects of hematuria on a 30% decline in eGFR during a five-year follow-up.

	cHR (95% CI)	*p*-Value	aHR (95% CI)	*p*-Value
Hematuria	2.063 (1.31, 3.24)	0.002	1.939 (1.23, 3.06)	0.005
Sex, Male	0.652 (0.48, 0.89)	0.008	0.714 (0.52, 0.98)	0.038
CCI	1.042 (1.01, 1.08)	0.019	1.030 (0.99, 1.07)	0.115
Albumin	0.776 (0.63, 0.96)	0.020	0.788 (0.63, 0.99)	0.037
NSAID	1.706 (1.24, 2.35)	0.001	1.572 (1.12, 2.21)	0.009

**Table 4 jcm-10-04345-t004:** Effects of hematuria on a 50% decline in eGFR during a five-year follow-up.

	cHR (95% CI)	*p*-Value	aHR (95% CI)	*p*-Value
Hematuria	3.839 (2.55, 5.77)	<0.001	3.447 (2.24, 5.30)	<0.001
Platelet	1.005 (1.00, 1.01)	0.007	1.005 (1.00, 1.01)	0.010
CRP	1.052 (1.01, 1.10)	0.027	1.043 (0.997, 1.09)	0.066
NSAID	1.584 (1.07, 2.35)	0.022	1.389 (0.92, 2.10)	0.119

**Table 5 jcm-10-04345-t005:** Sensitivity analysis.

	Renal Composite Outcomes		eGFR Decline 30%		eGFR Decline 50%	
	aHR (95% CI)	*p*-Value	aHR (95% CI)	*p*-Value	aHR (95% CI)	*p*-Value
Excluding eGFR < 60 (N = 205)						
Hematuria	2.913 (1.38, 6.13)	0.005	3.199 (1.58, 6.46)	0.001	2.773 (1.59, 4.83)	<0.001
Excluding UTI (N = 246)						
Hematuria	2.596 (1.04, 6.49)	0.041	2.330 (1.20, 4.54)	0.013	4.137 (2.26, 7.59)	<0.001
Excluding stone (N = 269)						
Hematuria	2.861 (1.50, 5.45)	0.001	2.930 (1.66, 5.16)	<0.001	4.633 (2.77, 7.74)	<0.001

## Data Availability

The data presented in this study are available on request from the corresponding author.

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
