# Peer review of "Association of Hematuria with Renal Progression and Survival in Patients Who Underwent Living Donor Liver Transplant"

_jcm, 2021, doi:10.3390/jcm10194345_

Round 1
Reviewer 1 Report
The authors design the study to determine the association of episodic or persistent hematuria following liver transplantation with long-term renal results. Patients with a live donor liver transplant were recruited and divided into two groups according to the discovery of microscopic or gross hematuria. In total, 295 patients underwent urinalysis for various reasons following the index date of surgery. Haematuria was detected in 100 patients (Group A) but was not observed in 195 patients (Group B). Compared to Arm B, there was a higher risk of renal progression in Arm A. In addition, patients who took non-steroidal anti-inflammatory drugs (NSAIDs) continuously for more than seven days within the six to 35 months prior to transplant surgery had a high risk of rapid renal progression. They concluded that the development of hematuria after surgery in patients who underwent living donor liver transplant and NSAIDs exposed before surgery were associated with worse long-term renal outcomes. Rationale is solid and plan of study is good. However, there are some concerns:
- What is the role of immune suppressants in liver transplants? The authors should explain in more detail on this subject whether will influence the urine test data.
- A more in-depth analysis of the weight of hematuria and proteinuria on the progression of kidney function should be undertaken. It can identify the one that is most important.
- In this study, hematuria had been consistently shown to predict renal outcomes in patients who received a living donor liver transplant. Because there are many causes of haematuria. Which one of these would be the most critical?
Author Response
Point to point answers to reviewer 1
Dear professor:
We thank you for your kindness and excellent recommendations for this manuscript.
- What is the role of immune suppressants in liver transplants? The authors should explain in more detail on this subject whether will influence the urine test data.
Response:
All patients had triple or double immunosuppressive regimen (calcineurin inhibitor (CNI)-tacrolimus, anti-metabolite- Mycophenolate mofetil or Mycophenolic acid, with or without steroid). CNI is a critical component of immunosuppressants and poses renal toxicity. We compared the average tacrolimus trough level and NSAID exposed during follow-up, there was no significant difference found between the two groups. Over suppression of the immune will increase the risk of urinary tract infection and malignancy. Infection and malignancy are the usual causes of hematuria.
- A more in-depth analysis of the weight of hematuria and proteinuria on the progression of kidney function should be undertaken. It can identify the one that is most important.
Response:
We appreciate the important question elicited. The severity of proteinuria associated with renal function decline, development of cardiovascular events, and overall mortality is well-established in CKD patients. In our patients, quantitative analysis of proteinuria (Proteinuria-creatinine ratio or 24 hours urine sample) was not monitored routinely and it was discussed in the limitations. However, as a clue, finding the underlying causes aggressively of hematuria and the following therapy will impact directly the outcomes.
- In this study, hematuria had been consistently shown to predict renal outcomes in patients who received a living donor liver transplant. Because there are many causes of haematuria. Which one of these would be the most critical?
Response:
Thanks!
Actually, it is difficult to define the most critical cause of hematuria
Furthermore, regular monitoring of a simple urinalysis and identifying the underlying causes aggressively or closed follow-up the renal function is the main purpose of this study.
Reviewer 2 Report
Thank you for inviting me to this interesting topic review. The authors investigated to determine the association of episodic or persistent hematuria after liver transplantation with long-term renal outcomes. This manuscript is well written and provides informative data in the liver transplantation community.
Page 2: The authors defined the index date as the date of achieving stable renal function after surgery, around three months after transplantation. Please, address the data related to renal function stability – such as a serial change in the creatinine levels – around these months.
Page 3: please, move the ethical sentence to the first paragraph in the 2. Material and Methods section.
Thank you.
Author Response
Point to point answers to reviewer 2
Dear professor:
We thank you for your kindness and excellent recommendations for this manuscript.
- Page 2: The authors defined the index date as the date of achieving stable renal function after surgery, around three months after transplantation. Please, address the data related to renal function stability – such as a serial change in the creatinine levels – around these months.
Response:
We appreciate the important question elicited.
In our patients who underwent living liver transplantation, 62% of patients encountered acute kidney injury (AKI) according to the KDIGO definition within 7 days after surgery. The series blood creatinine is mentioned in the table below.
|
Group A (Hematuria+) |
Group B (Hematuria-) |
P-value |
Creatinine at basaline |
1.24±0.79 |
1.22±0.82 |
0.898 |
Highest Cr during 7 days after surgery |
1.62±0.99 |
1.53±1.17 |
0.496 |
Creatinine at index date |
1.10 ± 0.44 |
1.07 ± 0.36 |
0.569 |
We also add the baseline creatinine in table 1. (page 4)
- Page 3: please, move the ethical sentence to the first paragraph in the 2. Material and Methods section..
Response:
Thanks! We had moved the ethical sentence to the first paragraph in the “Material and Methods section”. (Line 71, page2)